# Differences between the Sexes in the Relationship between Chronic Pain, Fatigue, and QuickDASH among Community-Dwelling Elderly People in Japan

**DOI:** 10.3390/healthcare9060630

**Published:** 2021-05-25

**Authors:** Satoshi Shimo, Yuta Sakamoto, Takashi Amari, Masaaki Chino, Rie Sakamoto, Masanori Nagai

**Affiliations:** 1Department of Occupational Therapy, Health Science University, Fujikawaguchiko, Yamanashi 401-0380, Japan; 2Department of Physical Therapy, Health Science University, Fujikawaguchiko, Yamanashi 401-0380, Japan; y.sakamoto@kenkoudai.ac.jp (Y.S.); takashi.amari@kenkoudai.ac.jp (T.A.); 3Yamanashi Research Institute Foundation, Kofu, Yamanashi 400-0031, Japan; chino-agrz@pref.yamanashi.lg.jp; 4Fuefuki City Council of Social Welfare, Fuefuki, Yamanashi 406-0822, Japan; chiiki04@fuefuki-shakyo.or.jp; 5Department of Welfare Psychology, Health Science University, Fujikawaguchiko, Yamanashi 401-0380, Japan; mnagai@kenkoudai.ac.jp

**Keywords:** chronic pain, sex differences, community-dwelling elderly people

## Abstract

Chronic pain and fatigue have negative effects on the health, ADL, work, and hobbies of the elderly. As the proportion of people 65 years of age and older in the population increases, chronic pain and disability research regarding this group is receiving more consideration. However, little empirical evidence of the association between chronic pain, fatigue, and physical disability between the sexes is available. This study investigated the association between chronic pain, fatigue, and instrumental activities of daily living among community-dwelling elderly people by sex in Japan. Concerning the presence of chronic pain, 61% of males and 78% of females reported chronic pain, indicating that many elderly people living in the community suffer from chronic pain and fatigue on a daily basis. The number of sites of chronic pain was higher in females than in males (*p* = 0.016), with more chronic pain in the knees (*p* < 0.001) and upper arms (*p* = 0.014). Regarding chronic pain, males showed a higher correlation with QuickDASH-DS (rs = 0.433, *p* = 0.017) and QuickDASH-SM (rs = 0.643, *p* = 0.018) than females. Furthermore, fatigue also showed a higher correlation with QuickDASH-W (rs = 0.531, *p* = 0.003) in males than in females. These results indicate that the association between chronic pain, fatigue, and QuickDASH differed between the sexes among community-dwelling elderly people in Japan. A better understanding of the risk factors for elderly chronic pain and fatigue among sexes will facilitate the development of elderly healthcare welfare and policies.

## 1. Introduction

Over recent decades, the average life expectancy has increased globally, reaching a global average of approximately 70 years in 2014 (6 years longer than in 1990) and reaching approximately 80 years in developed countries (compared to approximately 50 years in developed countries in the early 20th century) [1]. According to the World Health Organization, the global population of people aged ≥60 years was 600 million in 2000, and this is expected to rise to approximately two billion in 2050 [2]. Longer life is an incredibly valuable resource. For example, in high-income countries, including Japan, there is evidence that many people are rethinking rigid notions of what older age might consist of and are looking to spend these extra years in innovative ways, such as a new career, continuing education, or pursuing a neglected passion [3]. However, the proportion of people 65 years of age and older who report musculoskeletal pain and physical disability is high [4]. Various studies have been conducted on the epidemiology of chronic pain in the elderly, but no firm conclusions have yet been drawn, because the prevalence of chronic pain and its relationship to other variables, such as sex, vary by region [5]. In addition, little is known about the relationship between chronic pain and physical activity among males and females in community-dwelling elderly people [6,7,8,9].

It is widely acknowledged that males age at a faster rate biologically when compared to females [10]. On the other hand, Yu et al. identified a suite of RNA molecules in the bloodstream that are more likely to be elevated in women who developed chronic neck, shoulder, or back pain after a motor-vehicle accident [11]. Many of these RNA molecules are encoded by genes on the X chromosome, of which there were two copies in most women. Moreover, the type of pain hypersensitivity results from remarkably different pathways in male and female mice, with distinct immune cell types contributing to discomfort [12]. Future pain medications will be tailored to individuals, and sex will be a key factor in such personalized prescriptions. However, we still know little about the biological reasons for the longevity and morbidity gender gap.

Recent studies have demonstrated that the normative values for physical function with the QuickDASH questionnaire in males and females of the general population differ in other countries [13,14]. QuickDASH-related items concerning their regular daily activities (QuickDASH-DS), work (QuickDASH-W), and sports/music (QuickDASH-SM) module abilities enable conversion to a 100-point scale [15,16,17]. The QuickDASH questionnaire is composed of health assessment instruments, the reliability and validity of which have already been established [18,19]. However, it is still unclear whether QuickDASH scores among community-dwelling elderly people in Japan differ by sex. Previous studies on chronic pain in community-dwelling elderly people have shown that it is related to functional limitations and age [20,21,22]. Moreover, in many Western studies on chronic pain, being an immigrant and living in a socially problematic residential area had an impact on subjects [23,24]. However, in Japan, which has a small immigrant population and a well-developed social security system, the factors that influence chronic pain in community-dwelling elderly males and females are not clear. Many compelling publications have argued that sex and gender should be considered in preclinical, clinical, and population research [24,25].

At present, Japan has the highest life expectancy in the world, and by 2050, the average life expectancy is expected to exceed 80 years for males and 90 years for females [26]. With hyper-aging as a global issue, understanding the factors associated with chronic pain in community-dwelling elderly males and females is key to informing clinical management and minimizing disability [27]. In this study, we investigated whether factors affecting chronic pain among community-dwelling elderly people in Japan differ by sex.

## 2. Materials and Methods

### 2.1. Study Design

All the participants provided written informed consent prior to participation in the study, and the study conformed to the principles of the Declaration of Helsinki [28]. The study’s design was approved by the Research Ethics Committee of the Health Science University (approval number: 27-16). We sought the cooperation of the Yamanashi Research Institute Foundation and the Fuefuki City Council of Social Welfare in conducting this survey, undertaken between November 2014 and October 2015. Participation in the study was voluntary, and participants were recruited through introductions by older adults participating in local support projects, through municipalities, and through recommendations made by the staff members of the municipalities. Exclusion criteria comprised volunteers who were unable to answer the questionnaire due to reduced cognitive function, those unable to understand the objectives of the study, and those with sequelae or complications from diseases, such as (1) severe cerebrovascular disease and (2) diabetes mellitus, or (3) a movement disorder that could have serious effects on upper extremity function. For participants who had difficulty completing the written questionnaire due to the fact of reduced visual acuity or those that were unable to answer the written questionnaire because of fatigue midway through the procedure, we conducted face-to-face interviews and completed their questionnaires accordingly.

### 2.2. Outcome Measure

For the assessment, participants were required to respond to questions regarding the following: (1) the presence or absence of chronic pain and feelings of fatigue; (2) primary sites of chronic pain (pain for at least three months); (3) ADL assessment using the QuickDASH questionnaire; (4) their daily schedule (hours spent working and sleeping). Chronic pain was assessed according to pain sites and the visual analog scale (VAS). The VAS provides a high degree of resolution and is probably one of the most sensitive single-item measures for clinical pain research [29]. There is evidence for the validity and reliability of this scale when used by those who are given careful instruction and practice [30]. The degree of chronic pain was assessed on a VAS of 100 mm, with 0 mm representing “no pain” and 100 mm representing “unbearable pain” (pain VAS). For the pain site assessment, the sites were explained using a body chart, and participants described them freely. Fatigue was assessed using the VAS to evaluate the severity (fatigue VAS) [31,32]. A sample pie-chart time schedule for one day was presented to the participants, and they were asked to freely describe items pertaining to their daytime activities. The hours spent on daytime activities and sleeping were determined based on their responses to a question concerning working hours and specific content with regard to daytime activities (e.g., farm work and housework).

### 2.3. Statistical Analysis

The data are presented as means with standard deviations (SDs). Age, chronic pain, fatigue, QuickDASH, work hours, and sleeping hours were set as variables of the individual factors, and statistical analyses were performed using the unpaired *t*-test or Mann–Whitney U test. For statistical comparison of sites of chronic pain and individual daytime activities performed, Fisher’s exact test was used. For statistical analysis of QuickDASH, internal alignment in reliability was tested using Cronbach’s α reliability coefficient and standardized α to ensure the response results of this study. The variables for Spearman’s rank correlation coefficient and partial rank correlation coefficients were age, chronic pain, fatigue, QuickDASH, work hours, and sleep hours. Statistical analyses were performed using the R2.8.1 meta package and GraphPad Prism 6.07 (GraphPad software, San Diego, California, USA). The statistical software was used for all analyses at a significance level of *p* < 0.05.

## 3. Results

We included 111 community-dwelling older adults living in a rural area in central Japan, who were receiving no home care services (mean age, 75.0 ± 7.7 years; 36 males and 75 females), understood the study objectives, and agreed to participate in the study. The basic characteristics of the study participants and the male-to-female ratio are shown in Table 1. The mean chronic pain score of the respondents was 35.2 mm (SD = 26.4) for females and 23.6 mm (SD = 27.1) for males (*p* = 0.017). On the other hand, there was no statistically significant difference in the mean fatigue scores, QuickDASH-DS, QuickDASH-W, and QuickDASH-SM among males and females. Regarding the daytime schedule, there was a statistically significant difference between the mean working hours of males (5.5 h, SD = 3.6) and females (2.6 h, SD = 2.6) (*p* < 0.001). These results suggest that, among community-dwelling elderly people, females had a higher level of pain than that of males, but there was no difference in fatigue. Furthermore, the results suggest that males were much more active during daytime than females.

The numbers of chronic pain sites and the sites of chronic pain stratified by males and females are presented in Table 2 and Figure 1. The number of painful areas was high for both: 61% of males and 78% of females (*p* = 0.229). On the other hand, females were more likely than males to report chronic pain in multiple locations (*p* = 0.016). The most common sites of chronic pain were the lower back (with a prevalence in males of 31% and females of 32%), followed by the shoulders (with a prevalence in males of 25% and females of 23%). On the other hand, there was a statistically significant difference in the prevalence of knee pain, with 28% of females reporting it compared to 8% of males (*p* < 0.001). There was also a statistically significant difference in the prevalence of upper arm pain, with 7% of females compared to 0% of males (*p* = 0.014). Although there was no statistically significant difference, females had a higher prevalence of chronic pain in the elbows (4%), buttocks (1%), and thighs (3%) than males. These results suggest that the prevalence of chronic pain is high in both males and females among community-dwelling elderly people. Furthermore, the study suggested that the number of sites of chronic pain was higher in females than in males, with more chronic pain in the knees and upper arms.

Differences between the sexes in daytime activities are shown in Table 3. Out of all the male participants, 69% performed farm work each day, 11% performed indoor work, and 19% did not work. Out of the female participants, 35% performed farm work each day, 9% performed indoor work, 8% performed household work, and 48% did not work. Statistical analysis showed that more males than females were engaged in farm work during daytime (*p* < 0.001). On the other hand, more females than males did little or no activity during daytime (*p* < 0.001). These results suggest that males were more active during daytime and were more likely to be outdoors than females.

The internal consistency of QuickDASH is shown in Table 4. The internal consistency of the disability/symptom scale was high (Cronbach’s alpha total = 0.87; 0.92 of males and 0.84 of females). Cronbach’s alpha was 0.94 for the work (0.93 of males, 0.92 of females) and 0.95 for the sport/music scales (0.94 of males, 0.94 of females). The total correlations were substantial for all items, ranging from 0.85 to 0.97. Table 5 shows that there was no significant correlation between age and other variables of factors in males. However, in males, chronic pain results were strongly correlated with those of QuickDASH-SM (rs = 0.794, *p* < 0.001), QuickDASH-DS (rs = 0.601, *p* < 0.001), and fatigue (rs = 0.598, *p* < 0.001). QuickDASH-W results were also strongly correlated with those of QuickDASH-SM (rs = 0.801, *p* < 0.01) and QuickDASH-DS (rs = 0.709, *p* < 0.001) in males. However, there was no significant correlation between working and sleeping hours and other variables of factors in males. On the other hand, in females, while age correlated with those of work hours (rs = -0.442, *p* < 0.001). The QuickDASH-W results also correlated strongly with those of QuickDASH-DS (rs = 0.733, *p* < 0.001) and QuickDASH-SM (rs = 0.677, *p* < 0.001). However, in females, the correlation coefficients for fatigue (rs = 0.340, *p* = 0.003), QuickDASH-DS (rs = 0.435, *p* < 0.001), and QuickDASH-SM (rs = 0.336, *p* = 0.136) were all lower for chronic pain than in males. 

The partial correlation coefficients after correcting each item for chronic pain and fatigue are shown in Table 6. In chronic pain, males showed a higher correlation with QuickDASH-DS (rs = 0.433, *p* = 0.017) and QuickDASH-SM (rs = 0.643, *p* = 0.018) than females. In addition, for fatigue, males showed a higher correlation with QuickDASH-W (rs = 0.531, *p* = 0.003) than females. These results indicate that chronic pain and fatigue were strongly associated with ADL, work, and hobbies in community-dwelling elderly males. On the other hand, females showed a weaker correlation between chronic pain and other factors.

## 4. Discussion

The present study aimed to describe whether factors affecting chronic pain among community-dwelling elderly people in Japan differed between sexes. Concerning the presence of chronic pain reported by 73% of the participants (61% of males and 78% of females), this indicates that many community-dwelling elderly people in our study suffered from chronic pain on a daily basis. Consistent with other studies, females in this study were more likely to report overall chronic pain and more sites of chronic pain than males.

In this study, females presented with lower back and knee pain. In comparison, many males presented with shoulder and lower back pain. Previous studies have compared chronic pain measurements, such as the number of pain sites, with overall pain severity in relation to lower limb function (such as hip and knees) in the elderly [33,34,35]. These studies showed that there was an association between chronic pain and daytime activity in elderly people. Fethke reported that disabilities resulting from manual labor had different physical symptoms depending on the characteristics of the work [36]. In this study, the majority of the males engaged in farm work as a daytime activity, and common sites of pain were the shoulders and neck. This study was undertaken in a rural area where agricultural products, such as peaches and grapes, are actively cultivated. Therefore, ADL may be affected due to chronic pain in the shoulders and neck that results from overuse of the upper extremities while performing overhand farm work [37,38].

According to the literature data, the optimal Cronbach’s alpha values are >0.90. Values of 0.80–0.89 are considered good, 0.70–0.79 are acceptable, 0.60–0.69 are doubtful, 0.50–0.59 are weak, and <0.5 are unacceptable [39]. In this study, we found positive values supporting the internal consistency of QuickDASH. This study showed a strong correlation between QuickDASH items in males compared with females. Notably, the QuickDASH-SM results of males were correlated strongly with those of QuickDASH-W. Moreover, partial correlation showed that regarding chronic pain, there was a strong correlation between QuickDASH-DS and QuickDASH-SM in males. These results indicate that chronic pain control is especially important for males to maintain their work, hobbies, and living functions. Previous studies have shown that the quality of life of older adults who continue to work is satisfactory [40]. On the other hand, in this study, females were less active during daytime and more likely than males to have knee pain. The occupational engagement of community-dwelling older adults living independently is considered to have the potential to contribute to community and social planning for older adults. Their engagements provide interesting insights into older persons’ time use and occupational needs. Future studies may require consideration of daytime activities, such as farm-work-related movements and longitudinal studies, to perform ADL assessment during the busy farming seasons and off-seasons [41].

Fatigue was significantly related to poorer physical performance, slower walking speed, lower mobility, and disability in IADLs by cross-sectional studies [42,43]. Battalio reported that females with reduced physical activity had depressive symptoms, such as general fatigue and feelings of fatigue, at a significantly higher rate [44]. However, in this study, fatigue was higher in males than in females, which diverges from previous reports [45,46]. Interestingly, fatigue in males was strongly correlated with QuickDASH-W, suggesting an interaction between the factors. The strength of this study was that it used partial correlation. These results may indicate that a certain type of chronic pain and fatigue sensitivity results from different pathways in males and females [10,11,12]. With regard to the reporting of fatigue, a study on the differences between the sexes found that among females, only biological complaints and psychosocial problems were related to fatigue, whereas in males, having a severe handicap and severe chronic complaints were related to fatigue [47]. Additional longitudinal studies are needed to elucidate gender-specific differences but should also investigate the multiple dimensions of fatigue and its long-term effect on functioning [48].

This study had several limitations. First, the impact of the participants’ medical histories on chronic pain was not fully examined [49,50]. Second, the sample size was too small to perform a multiple regression analysis. Third, participants were recruited through introductions from older adults participating in local community support activities, through municipality recruitment, or through recommendations by municipality staff members. Therefore, further studies will be needed in populations of older adults with different lifestyles who stay at home and those who have reduced opportunities to leave their homes [40,51,52]. However, despite the limitations, the study has a major strength: the results of this study suggest that the relationship between chronic pain and other factors differs by sex among community-dwelling elderly people in Japan. Keeping in mind the limitations of the present study, we plan to perform further research with a larger sample size in order to generalize the findings.

## 5. Conclusions

We examined whether factors affecting chronic pain among community-dwelling elderly people in Japan differed by sex. Concerning the presence of chronic pain, 61% of males and 78% of females reported chronic pain, indicating that many elderly people living in the community suffer from chronic pain and fatigue on a daily basis. The number of sites of chronic pain was higher in females than in males, with more chronic pain in the knees and upper arms. Regarding chronic pain, males showed a higher correlation with QuickDASH-DS and QuickDASH-SM than females. Furthermore, fatigue also showed a higher correlation with QuickDASH-W in males than in females. Public health policies that bring older males into productive engagement outside the home may thus improve their decreased chronic pain and generate benefits for communities. These results indicate that the association between chronic pain, fatigue, and QuickDASH differed between the sexes among community-dwelling elderly people in Japan. A better understanding of the risk factors for elderly chronic pain and fatigue among sexes will facilitate the development of elderly healthcare welfare and policies.

## Figures and Tables

**Figure 1 healthcare-09-00630-f001:**
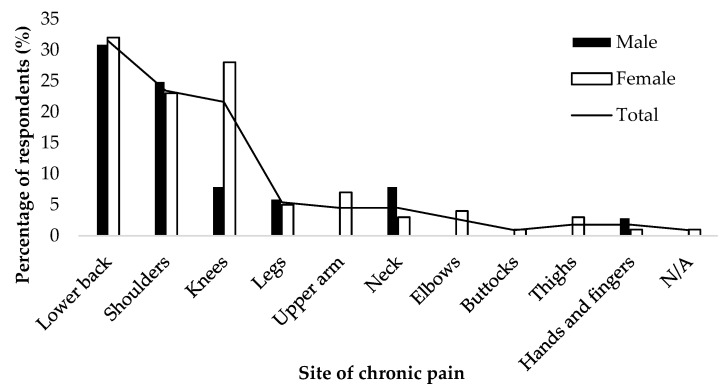
Percentage of chronic pain according to anatomical sites.

**Table 1 healthcare-09-00630-t001:** Basic characteristics of the study participants by males and females.

	Male (*n* = 36)	Female (*n* = 75)	Total (*n* = 111)	*p*-Value
Age, years (SD)	71 (6.7)	77 (7.3)	75 (7.7)	<0.001
Chronic pain, mm (SD)	23.6 (27.1)	35.2 (26.4)	31.5 (27.6)	0.017
Fatigue, mm (SD)	16.1 (24.6)	13 (21.7)	13.9 (23.4)	0.568
QDASH-DS (SD)	11.4 (13.7)	11.7 (14.3)	11.6 (14.5)	0.959
QDASH-W (SD)	9.9 (16.6)	10.5 (14.9)	10.3 (16.3)	0.868
QDASH-SM (SD)	8.5 (14.4)	4.5 (9.7)	6.1 (13.7)	0.225
Work hours (SD)	5.5 (3.6)	2.6 (2.6)	3.6 (3.2)	<0.001
Sleeping hours (SD)	6.9 (1.3)	7 (1.7)	6.9 (1.4)	0.739

QDASH-DS, QuickDASH disability/symptom; QDASH-W, QuickDASH work; QDASH-SM, QuickDASH sports/music; SD, standard deviations; VAS, visual analog scale.

**Table 2 healthcare-09-00630-t002:** Prevalence of chronic pain in different body regions between males and females.

	Chronic Pain Site (%)		
	Male	Female	*p*-Value
Number of pain sites			
Only one site	18 (50)	40 (53)	0.777
More than one	4 (11)	19 (25)	0.016
Total	22 (61)	59 (78)	0.229
Site of chronic pain			
Lower back	11 (31)	24 (32)	1
Shoulders	9 (25)	17 (23)	0.869
Knees	3 (8)	21 (28)	<0.001
Legs	2 (6)	4 (5)	1
Upper arm	0 (0)	5 (7)	0.014
Neck	3 (8)	2 (3)	0.213
Elbows	0 (0)	3 (4)	0.121
Buttocks	0 (0)	1 (1)	1
Thighs	0 (0)	2 (3)	0.621
Hands and fingers	1 (3)	1 (1)	0.621
N/A	0 (0)	1 (1)	1

N/A, not applicable.

**Table 3 healthcare-09-00630-t003:** Characteristics of the individual daytime activities of the study subjects between males and females.

	Subjects (%)		
Daytime Activities	Male	Female	*p*-Value
Farm work	25 (69)	26 (35)	<0.001
Indoor work	4 (11)	7 (9)	0.814
Household work	0 (0)	6 (8)	0.007
N/A	7 (19)	36 (48)	<0.001

N/A, not applicable.

**Table 4 healthcare-09-00630-t004:** Internal consistency of QuickDASH between males and females.

	Cronbach’s α Reliability Coefficient (Range)		
	Male	Female	Total
QDASH-DS	0.92 (0.90–0.91)	0.84 (0.81–0.85)	0.87 (0.85–0.88)
QDASH-W	0.93 (0.89–0.92)	0.92 (0.88–0.91)	0.94 (0.90–0.94)
QDASH-SM	0.94 (0.91–0.94)	0.94 (0.89–0.96)	0.95 (0.88–0.97)

QDASH-DS, QuickDASH disability/symptom; QDASH-W, QuickDASH work; QDASH-SM, QuickDASH sports/music.

**Table 5 healthcare-09-00630-t005:** Coefficients of correlation between chronic pain, fatigue, QuickDASH, working hours, and sleeping hours in males and females. Spearman’s rank correlation coefficient (upper) and *p*-value (lower) between the chronic pain, fatigue and QuickDASH characteristics for males and females.

		Chronic Pain	Fatigue	QDASH-DS	QDASH-W	QDASH-SM	Work Hours	Sleeping Hours
Age	Male	−0.336	−0.075	−0.179	−0.096	−0.364	−0.284	0.130
		0.052	0.672	0.344	0.612	0.199	0.121	0.472
	Female	−0.198	−0.148	0.059	0.167	0.269	−0.442	0.143
		0.092	0.212	0.642	0.214	0.238	0.001	0.230
Chronic pain	Male	1	0.598	0.601	0.411	0.794	0.130	−0.244
			0.001	0.001	0.022	0.001	0.479	0.164
	Female	-	0.340	0.435	0.267	0.336	0.125	−0.021
			0.003	0.001	0.041	0.136	0.333	0.859
Fatigue	Male	-	1	0.477	0.638	0.665	−0.084	−0.006
				0.007	0.001	0.008	0.647	0.974
	Female	-	-	0.302	0.467	0.030	0.070	−0.123
				0.013	0.001	0.896	0.589	0.296
QDASH-DS	Male	-	-	1	0.709	0.597	0.217	−0.240
					0.001	0.034	0.250	0.311
	Female	-	-	-	0.733	0.397	0.071	−0.118
					0.001	0.103	0.603	0.347
QDASH-W	Male	-	-	-	1	0.801	0.011	−0.188
						0.001	0.955	0.311
	Female	-	-	-	-	0.677	0.066	−0.024
						0.001	0.657	0.493
QDASH-SM	Male	-	-	-	-	1	−0.109	−0.426
							0.709	0.130
	Female	-	-	-	-	-	−0.161	0.163
							0.522	0.493
Work Hours	Male	-	-	-	-	-	1	−0.344
								0.054
	Female	-	-	-	-	-	-	−0.117
								0.371

QDASH-DS, QuickDASH disability/symptom; QDASH-W, QuickDASH work; QDASH-SM, QuickDASH sports/music.

**Table 6 healthcare-09-00630-t006:** Partial correlation coefficient of chronic pain and fatigue in males and females (both after adjusting for chronic pain and fatigue). Partial correlation coefficients (upper) and *p*-value (lower) of chronic pain and fatigue in males and females (both after adjusted pain and fatigue).

	Age	QDASH-DS	QDASH-W	QDASH-SM	Work Hours	Sleeping Hours
Chronic pain						
Male	−0.381	0.433	−0.095	0.643	0.236	−0.319
	0.029	0.017	0.614	0.018	0.201	0.070
Female	−0.158	0.364	0.128	0.342	0.028	−0.022
	0.184	0.003	0.339	0.140	0.406	0.849
Fatigue						
Male	0.206	0.137	0.531	0.339	−0.215	0.213
	0.250	0.471	0.003	0.258	0.244	0.235
Female	−0.084	0.170	0.262	0.296	0.108	−0.124
	0.484	0.173	0.001	0.757	0.828	0.297

QDASH-DS, QuickDASH disability/symptom; QDASH-W, QuickDASH work; QDASH-SM, QuickDASH sports/music.

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
