# Peer review of "Differences between the Sexes in the Relationship between Chronic Pain, Fatigue, and QuickDASH among Community-Dwelling Elderly People in Japan"

_healthcare, 2021, doi:10.3390/healthcare9060630_

Round 1
Reviewer 1 Report
This manuscript addresses the important topic of chronic pain in elderly, a not well understood nor investigated topic. The authors have chosen to investigate the differences of chronic pain in older men and women in a rural setting in Japan. An interesting topic, However maybe a bit too focused. I wonder if in the retrieved data more insights in pain in older people can be analyzed.
The introduction is well written and is informative yet the it stays a bit superficial.
I miss a very recent review on the topic of pain in elderly that can be helpful in the general introduction --> Zhao Y, Zhang Z, Guo S, Feng B, Zhao X, Wang X, Wang Y. Bibliometric Analysis of Research Articles on Pain in the Elderly Published from 2000 to 2019. J Pain Res. 2021;14:1007-1025
https://doi.org/10.2147/JPR.S283732
Furthermore why is it of interest to look at differences between males and females regarding chronic pain? What is still unknown and why should it be investigated? In other words what are the potential consequences of a lack of insight in the difference between men and women?The introduction can be improved by more in-depth information regarding this problem.
In the method section, line 70-73 the sentence ‘We included 111 community-dwelling older adults living in a rural area in central Japan, who were receiving no home care services (mean age, 75.0 ± 7.7 years; 36 males and 75 females), understood the study objectives, and agreed to participate in the study (Table 1). ‘ ïƒ should be replaced to the result section
Furthermore in the method section, regarding the paragraph on statistical analyses; The authors state in the method section, 2.2. outcome measures, line 100; The Quick-DASH questionnaire is composed of health assessment instruments, the reliability and validity of which have already been established [24,25] ïƒ regarding this statement it is not clear why the authors decided to the following (line 112-113): “For statistical analysis of QuickDASH, internal alignment in reliability was tested using Cronbach’s α reliability coefficient and standardized α.”. Please elaborate on this.
In the discussion section it is stated in line 341 that it has been tried to perform a multiple regression analysis. This is, However, not mentioned in the statistical analyses section. Why not? And why is the result of this analyses not mentioned in the result section. It is my opinion that it is stronger to show the full regression model, although probably all factors are not significant. The authors write about this in line 341 the following; ‘the sample size was too small to perform a multiple regression analysis, so it did not become common’ ïƒ it is not clear to me what the authors mean with ‘not become common’ ?
The result section is in general well written although in most parts the text can be more condensed and focused on the main and statistical significant findings from the analysis. The remaining results can be found in the tables.
The discussion is a bit weak. Please try to elaborate more on the findings in relation to the current literature and specifically the lack of literature on the specific topic you investigated. What is new ?!
The conclusion is not well written. The authors just repeat some of the main results. This must be improved.
Author Response
Response to Reviewer 1 Comments
Point 1: The introduction is well written and is informative yet the it stays a bit superficial. I miss a very recent review on the topic of pain in elderly that can be helpful in the general introduction --> Zhao Y, Zhang Z, Guo S, Feng B, Zhao X, Wang X, Wang Y. Bibliometric Analysis of Research Articles on Pain in the Elderly Published from 2000 to 2019. J Pain Res. 2021;14:1007-1025 https://doi.org/10.2147/JPR.S283732
Response 1: Thank you for your suggestion. We have revised the introduction section.
“However, the proportion of people 65 years of age and older who report musculoskeletal pain and physical disability is high [4].” Reference 4 was added.
Point 2: Furthermore why is it of interest to look at differences between males and females regarding chronic pain? What is still unknown and why should it be investigated? In other words what are the potential consequences of a lack of insight in the difference between men and women? The introduction can be improved by more in-depth information regarding this problem.
Response 2: Thank you for your suggestion. We have revised the introduction section.
“On the other hand, Yu et al. has identified a suite of RNA molecules in the blood-stream that are more likely to be elevated in women who develop chronic neck, shoulder or back pain after a motor-vehicle accident [11]. Many of these RNA mole-cules are encoded by genes on the X chromosome, of which there were two copies in most women. Moreover, kind of pain hypersensitivity results from remarkably dif-ferent pathways in male and female mice, with distinct immune-cell types contributing to discomfort [12]. Future pain medications will be tailored to individuals and that sex will be a key factor in those personalized prescriptions. However, we still know little about the biological reasons for the longevity and morbidity gender gap.” Reference 11, 12 were added.
Point 3: In the method section, line 70-73 the sentence ‘We included 111 community-dwelling older adults living in a rural area in central Japan, who were receiving no home care services (mean age, 75.0 ± 7.7 years; 36 males and 75 females), understood the study objectives, and agreed to participate in the study (Table 1). ‘ ïƒ should be replaced to the result section
Response 3: To address this comment, we have replaced to the results section
Point 4: Furthermore in the method section, regarding the paragraph on statistical analyses; The authors state in the method section, 2.2. outcome measures, line 100; The Quick-DASH questionnaire is composed of health assessment instruments, the reliability and validity of which have already been established [24,25] ïƒ regarding this statement it is not clear why the authors decided to the following (line 112-113): “For statistical analysis of QuickDASH, internal alignment in reliability was tested using Cronbach’s α reliability coefficient and standardized α.”. Please elaborate on this.
Response 4: Thank you for your suggestion. We have revised the statistical analysis section and the discussion section, added reference 39.
“For statistical analysis of QuickDASH, internal alignment in reliability was tested using Cronbach’s α reliability coefficient and standardized α to ensure the response results of this study.”
“According to the literature data, the optimal Cronbach’s alpha values are >0.90. Values of 0.80–0.89 are considered good, 0.70–0.79 are acceptable, 0.60–0.69 are doubtful, 0.50–0.59 are weak, and <0.5 are unacceptable [39]. In this study, we found positive values supporting the internal consistency of QuickDASH.”
Point 5: In the discussion section it is stated in line 341 that it has been tried to perform a multiple regression analysis. This is, However, not mentioned in the statistical analyses section. Why not? And why is the result of this analyses not mentioned in the result section. It is my opinion that it is stronger to show the full regression model, although probably all factors are not significant. The authors write about this in line 341 the following; ‘the sample size was too small to perform a multiple regression analysis, so it did not become common’ ïƒ it is not clear to me what the authors mean with ‘not become common’ ?
Response 5: Thank you for your comment, we have revised the sentence.
“Second, the sample size was too small to perform a multiple regression analysis.”
Point 6: The result section is in general well written although in most parts the text can be more condensed and focused on the main and statistical significant findings from the analysis. The remaining results can be found in the tables.
Response 6: To address this comment, we has been deleted.
First paragraph: “(p = 0.568)”," (p = 0.959) ", " (p = 0.868) ", " (p = 0.225) " and "On the other hand, for sleep time, there was no significant difference between males (6.9 hours, SD = 1.3) and females (7 hours, SD = 1.7) (p = 0.739). "
Second paragraph: “Concerning the presence of chronic pain, 61% of males and 78% of females reported chronic pain, indicating that many elderly people living in the community suffer from chronic pain and fatigue on a daily basis (p = 0.229). “
Point 7: The discussion is a bit weak. Please try to elaborate more on the findings in relation to the current literature and specifically the lack of literature on the specific topic you investigated. What is new ?!
Response 7: Thank you for your suggestion. We have revised the discussion section.
“The strength of this study was that it used partial correlation. These results may indicate that a certain type of chronic pain and fatigue sensitivity results from different pathways in males and females [10–12].”
Point 8: The conclusion is not well written. The authors just repeat some of the main results. This must be improved
Response 8: To address your comment, we have revised the conclusions section
“Public health policies that bring older males into productive engagement outside the home may thus improve their decreased chronic pain and generate benefits for communities.”
Thank you for your comments and suggestions.

Reviewer 2 Report
Your work is very good. But I go to add some comments about your paper for improve this manuscript.
INTRODUCTION:
The introduction is correct, follow a order and all cites are correct. Congratulations.
METHODS:
In Helsinki Declaration, you should add the reference and cite about this line in your manuscript.
RESULTS:
You should improve the results part.
It is advisable to add a figure to the results section, for example in the significant variable "Site of chronic pain".
Table 3, can't have got "-" in title, daytime activities.
All tables of the manuscript, should appear the same format.
All tables in the manuscript must appear in the same format. Especially dashes and line shapes.
DISCUSSION:
It's correct, and respond all questions about your work.
Author Response
Response to Reviewer 2 Comments
Your work is very good. But I go to add some comments about your paper for improve this manuscript.
INTRODUCTION:
Point 1: The introduction is correct, follow a order and all cites are correct. Congratulations.
Response 1: Thank you for your comment.
METHODS:
Point 2: In Helsinki Declaration, you should add the reference and cite about this line in your manuscript.
Response 2: Thank you for your suggestion. We have added the reference 20. World Medical Association declaration of Helsinki: Ethical principles for medical research involving human subjects. JAMA - J. Am. Med. Assoc. 2013, 310, 2191–2194.
RESULTS:
You should improve the results part.
Point 3: It is advisable to add a figure to the results section, for example in the significant variable "Site of chronic pain".
Response 3: Thank you for your suggestion. We have added Figure 1. Percentage of chronic pain according to anatomical sites.
Point 4: Table 3, can't have got "-" in title, daytime activities.
Response 4: Per your suggestion, Table 3 "-" in title, daytime activities has been deleted.
Point 5: All tables of the manuscript, should appear the same format. All tables in the manuscript must appear in the same format. Especially dashes and line shapes.
Response 5: Per your suggestion, we have revised all tables in the same format.
DISCUSSION:
Point 6: It's correct, and respond all questions about your work.
Response 6: Thank you for your comment.
Thank you for your comments and suggestions.

Reviewer 3 Report
The manuscript aims to examine the differences between the sexes in the relationship between chronic pain, fatigue, and QuickDASH among community dwelling elderly in Japan. My main concern is that the male participants selected were significantly younger than female participants in the study, which could be a major confounder in the analyses presented in the manuscript. Also, the study sample size (total N of 111) is very small for an observational study for this kind, which means it is difficult to perform analyses to account for confounders, or to draw meaningful conclusions. Additionally, recruitment bias may also limit the generalisability of the study results to other Japanese populations – although this has already been acknowledged by the authors in the discussion as a study limitation.
Main comments:
- The background and the use of QuickDASH scores for monitoring chronic pain should be introduced in the introduction section.
- It is widely acknowledged that men age at a faster rate biologically when compared to women, and this should be discussed accordingly (e.g. Eitaro Nakamura, Kenji Miyao, Sex Differences in Human Biological Aging, The Journals of Gerontology: Series A, Volume 63, Issue 9, September 2008, Pages 936–944, https://doi.org/10.1093/gerona/63.9.936)
- It is interesting that QuickDASH scores were found more strongly associated to chronic pain and fatigue in men than in women in the study. Do the authors think QuickDASH scores are more reliable and informative instruments in elderly men than in elderly women?
- Line 124-125: Do the authors mean “between male and female” or “among male”? Please check the grammar.
- Table 1. Sample number (N) for male and female should be indicated. I cannot find information on “male-to-female ratio” from the table, despite the table legend.
Author Response
Response to Reviewer 3 Comments
Main comments:
Point 1: The background and the use of QuickDASH scores for monitoring chronic pain should be introduced in the introduction section.
Response 1: To address this comment, we have revised the introduction section.
“Recent studies have demonstrated that the normative values for physical function with the QuickDASH questionnaire in males and females of the general population differ in other countries [13,14]. QuickDASH-related items concerning their regular daily activities (QuickDASH-DS), work (QuickDASH-W), and sports/music (QuickDASH-SM) module abilities enable conversion to a 100-point scale [15–17]. The QuickDASH questionnaire is composed of health assessment instruments, the reliability and validity of which have already been established [18,19]. However, it is still unclear whether QuickDASH scores among community-dwelling elderly people in Japan differ by sex.”
Point 2: It is widely acknowledged that men age at a faster rate biologically when compared to women, and this should be discussed accordingly (e.g. Eitaro Nakamura, Kenji Miyao, Sex Differences in Human Biological Aging, The Journals of Gerontology: Series A, Volume 63, Issue 9, September 2008, Pages 936–944, https://doi.org/10.1093/gerona/63.9.936)
Response 2: To address this comment, we have revised the introduction section.
“It is widely acknowledged that males age at a faster rate biologically when compared to females [10]. On the other hand, Yu et al. identified a suite of RNA molecules in the bloodstream that are more likely to be elevated in women who developed chronic neck, shoulder, or back pain after a motor-vehicle accidents [11]. Many of these RNA molecules are encoded by genes on the X chromosome, of which there were two copies in most women. Moreover, the type of pain hypersensitivity results from remarkably different pathways in male and female mice, with distinct immune cell types contributing to discomfort [12]. Future pain medications will be tailored to individuals, and sex will be a key factor in such personalized prescriptions. However, we still know little about the biological reasons for the longevity and morbidity gender gap”
Point 3: It is interesting that QuickDASH scores were found more strongly associated to chronic pain and fatigue in men than in women in the study. Do the authors think QuickDASH scores are more reliable and informative instruments in elderly men than in elderly women?
Response 3: We appreciate the reviewer's comment on this point. It is very important point. “These results may indicate that a certain type of chronic pain and fatigue sensitivity results from different pathways in males and females [10–12].”, “Public health policies that bring older males into productive engagement outside the home may thus improve their decreased chronic pain and generate benefits for communities.”. We have revised the discussion and conclusions section.
Point 4: Line 124-125: Do the authors mean “between male and female” or “among male”? Please check the grammar.
Response 4: Thank you for your suggestion. We added “and females”.
Point 5: Table 1. Sample number (N) for male and female should be indicated. I cannot find information on “male-to-female ratio” from the table, despite the table legend.
Response 5: Thank you for your suggestion. We have revised Table 1 title, added sample number (N).
Thank you for your comments and suggestions.
